# Tuberculosis patients face high treatment support costs in Colombia, 2021

Oscar Andrés Cruz Martínez[1], Ingrid García[2], Gloria Mercedes Puerto[3], Nelson J. Alvis-Zakzuk[4,5,6]*, Martha Patricia López[2], Juan Carlos Moreno Cubides[2], Ángela María Sánchez Salazar[7], Julián Trujillo Trujillo[8,9], Claudia Marcela Castro-Osorio[3], Vivian Vanessa Rubio[3], Carlos Castañeda-Orjuela[10], Ernesto Montoro[11], Peter Nguhiu[4], Inés García Baena[4]

1 Programa Nacional para el Control de la Tuberculosis, Ministerio de Salud y Protección Social, Bogotá, D. C., Colombia, 2 Área Prevención y Control de Enfermedades CDE, Organización Panamericana de la Salud/ Organización Mundial de la Salud, Bogotá, D.C., Colombia, 3 Instituto Nacional de Salud, Red Nacional de Investigación, Innovación y Gestión del Conocimiento de TB en Colombia, Grupo de Micobacterias, Dirección de Investigación en Salud Pública, Bogotá, D.C., Colombia, 4 Global TB Programme, World Health Organization, Geneva, Switzerland, 5 Department of Health Sciences, Universidad de la Costa, Barranquilla, Colombia, 6 Laboratory of Causal Inference in Epidemiology (LINCE-USP), School of Public Health, Postgraduate Program in Epidemiology, University of São Paulo, São Paulo, Brazil, 7 Dirección de Epidemiologia y Demografía, Ministerio de Salud y Protección Social, Bogotá, D.C., Colombia, 8 Grupo Emergentes Reemergentes y Desatendidas, Ministerio de Salud y Protección Social, Bogotá, D.C., Colombia, 9 Universidad Nacional Abierta y a Distancia UNAD, Bogotá, D.C., Colombia, 10 Instituto Nacional de Salud, Observatorio Nacional de Salud, Bogotá, D.C., Colombia, 11 Departamento CDE, Unidad de VIH, Hepatitis, Tuberculosis e ITS, Organización Panamericana de la Salud/Organización Mundial, Washington, DC, Estados Unidos de América

* alviszakzuk@gmail.com

**Data Availability Statement:** Survey data sets contain privacy-sensitive information including participants' individual and household income that formed a core part of the analysis. Even though we

## Abstract

### Objective

To estimate the baseline to measure one of the three indicators of the World Health Organization (WHO) End TB strategy (2015–2035), measure the costs incurred by patients affected by tuberculosis (TB) during a treatment episode and estimate the proportion of households facing catastrophic costs (CC) and associated risk factors, in Colombia, 2021.

### Material and methods

A nationally representative cross-sectional survey was conducted among participants on TB treatment in Colombia, using telephone interviews due to the exceptional context of the COVID-19 pandemic. The survey collected household costs (direct [medical and non-medical out-of-pocket expenses] and indirect) over an episode of TB, loss of time, coping measures, self-reported income, and asset ownership. Total costs were expressed as a proportion of annual household income and analyzed for risk factors of CC (defined as costs above 20% annual household income).

### Results

The proportion of TB-affected households incurring in costs above 20% annual household income (CC) was 51.7% (95%CI: 45.4–58.0) overall, 51.3% (95%CI: 44.9–57.7) among

**Funding:** YES. the following institution funded this study: World Health Organization Panamerican Health Organization Ministerio de Salud y Protección Social Instituto Nacional de Salud The funders had no role in study design, data collection and analysis, decision to publish, or preparation of the manuscript.

**Competing interests:** The authors have declared that no competing interests exist.

patients with drug-sensitive (DS) TB, and 65.0% (95%CI: 48.0–82.0) among drug-resistant (DR). The average patient cost of a TB case in Colombia was $1,218 (95%CI 1,106–1,330) including $860.9 (95%CI 776.1–945.7) for non-medical costs, $339 (95%CI 257–421) for the indirect costs, and $18.1 (95%CI 11.9–24.4) for the medical costs. The factors that influenced the probability of facing CC were income quintile, job loss, DR-TB patient, and TB type.

## Conclusion

Main cost drivers for CC were non-medical out-of-pocket expenses and income loss (indirect costs). Current social protection programs ought to be expanded to mitigate the proportion of TB-affected households facing CC in Colombia, especially those with lower income levels.

## Introduction

Tuberculosis (TB) remains a national and global public health priority and in 2021, it was the second leading infectious disease killer worldwide after COVID-19 [1,2]. The World Health Organization (WHO) reported, in 2021, a global TB incidence rate of 134 cases per 100,000 persons, the first time in more than two decades that the incidence rate had risen on an annual basis [3]. In 2021, the National Programme for Prevention and Control of Tuberculosis (PNPCT, acronym in Spanish) of Colombia notified 13,659 (that is, new and relapse) TB giving a notification rate of 26.5 cases per 100,000 persons and showing an increase of 16% compared to the previous year's notification rate of 22.8 [4]. The WHO estimates that these notifications represented 67% of the estimated annual 21,000 (17 000–26 000) falling ill with TB (incidence number) in Colombia [5]; with a considerable gap in the successful detection of all new TB cases [5]. Moreover, the PNPCT reports that only 71% of cases under treatment in 2020 achieved successful treatment outcomes (target of 90%), and 10% of cases were lost to follow-up during the treatment process [6].

Owing to the impact of TB on the population, the WHO, as part of its End TB strategy, established the goal of eliminating catastrophic costs (CC) for all TB-affected households, defined as costs representing more than 20% of household income [7]. In 2023, the WHO estimated that 49% (95% confidence interval [CI]: 37–61%) of households affected by TB experienced CC, as shown in the 29 national surveys taken between 2016–2022 [2,8]. CC can vary between countries depending on access to care, health financing policies, coverage, and health plans offered to the population [9]. Of the 29 countries who measured their CC baseline, the percentage varied from 13% (95% confidence interval [CI]: 10–17%) in El Salvador to 92% (95% CI: 86–97%) in the Solomon Islands [8,10].

With this in mind, the Ministry of Health and Social Protection (MSPS, in Spanish) in accordance with international policies, adopted the National Strategic Plan: Colombia towards the End of TB 2016–2025 [11], which intends to progressively reduce the incidence, mortality, and number of households experiencing CC due to tuberculosis. This last goal implies the identification of a baseline of TB-affected households experiencing CC. For this reason, our study aims to estimate the proportion of CC, the costs of a TB episode, and the factors associated with the probability of facing CC.

## Materials and methods

### Study setting

Colombia is a South American low-and middle-income country, with an estimated 51.5 million inhabitants in 2021 [12] of whom 2.4% are estimated to be living below the international poverty threshold of 1.9 PPP$ per day [13,14]. The country is administratively divided into the capital district (Bogotá, D.C.) and 32 other departments, which are further subdivided into 1,122 municipalities (11). The country's health care system is estimated to reach over 99% of the population in 2022 [14]. This system is composed of a contributive regime, which covers people who are able to pay; a subsidized regime, which covers those who are not able to pay; and a special one, which covers teachers, armed forces, and national police. Additionally, there are complimentary private healthcare and prepaid medicine plans, frequently used by wealthier factions of the population [15].

### Study design

A national representative, stratified, cluster sampled cross-sectional survey was conducted to estimate the incidence of CC among TB-affected households in 2021. The design of the study, the objectives, and its methodology were guided by the World Health Organization's Handbook on National TB patient cost surveys [16], but with unique features introduced in the Colombian survey such as the exclusive use of computer assisted telephone interview technique and the addition of country contextual questions on UHC service coverage into the data collection instrument. The key aspects of the study's design are detailed below.

### Population

The study population was composed of people notified within the PNPCT with either drug-sensitive (DS-TB) or drug-resistant TB (DR-TB) and undergoing TB treatment at the time of the study for at least 14 days in either intensive or continuation phase. Since the survey questionnaire relied on self-reported data and required recollection of recent costs, those who were not currently on treatment for TB, and those with any cognitive, auditory, or language impairments were excluded from the study. Additionally in line with the protocol's institutional review board recommendation, those who were homeless, incarcerated, or those younger than 18 years were all deemed ineligible and excluded for this study. From the most recent complete notification list at the PNPCT, 5,656 individuals were eligible and on treatment between January 1 and December 31 of 2020, distributed over 544 municipalities in all 32 departments of the country [17].

### Procedures

**Sampling.**   The primary sampling unit was the municipality in which facilities were providing TB treatment, while the secondary sampling unit (and the unit of analysis) was the individual on treatment within the network of TB treatment facilities in a sampled municipality.

To prepare the stratified sampling frame, municipalities were classified in two initial groups (Stratum 1: municipalities with 30 or more people on treatment for TB; Stratum 2: municipalities with less than 30 people on treatment for TB). The first stratum contained 33 (6%) municipalities and included 67% of all persons eligible for the study; the second stratum contained 511 (94% of municipalities) with the remaining 33% of eligible persons. Notably, 79% of municipalities had <6 persons on treatment which posed a challenge for the probability proportional to size sampling of these clusters and a threat of low sample yield. Therefore,

municipalities in the second stratum were grouped, based on geographic proximity within the same region, to give 54 clusters with nine municipalities and 33 TB patients on average.

The sample size was calculated to estimate a national level CC proportion of 50% with an absolute precision of +/- 2.5 percentage points, while accounting for a design effect of 1.2, and assuming a sample non-response rate of 20%. The sample size calculated was 1,313 people distributed proportionate to each stratum's eligible number of persons. Twenty-seven clusters were then randomly sampled with probability proportional to the number of persons eligible.

**Survey and data collection.** The survey was conducted between 13th of July and 5th of August in 2021. Eligible participants were selected from each cluster and invited to take part in a phone survey. The phone method had not been implemented in other countries, but due to the COVID-19 pandemic it was chosen as the best way for data collection. This required that the generic survey questionnaire developed by WHO be first adapted to Colombian contextual factors such as income, household assets, health system, and others [16], and thereafter piloted. The pilot survey was performed 14 times in person and 8 times over the phone in five departments of Colombia between November and December of 2020. After the pilot, the survey tool was finalized and coded onto electronic format using an Open Data Kit (ODK) backed, web-based data capture platform maintained by ONA (Ona Systems Inc.) [18].

The survey team comprised of 20 trained interviewers, 8 supervisors, a data manager, and 24 support coordinators and a technical advisory group. The interviewers were provided with mobile phones for exclusive use in the survey and used them to carry out the interviews. The participants' telephone numbers were obtained from the PNPCT electronic case-based register. The interviewers followed an approved script to conduct pre-interview introduction, to request for informed consent, to capture an audio recording of consent and to perform the interviews to avoid bias. In process quality control was handled by the supervisors who reviewed the data captured [16,17].

The questionnaire [19] consisted of four parts: 1. Personal information for TB patient obtained from the electronic registry of the PNPCT and from the scanned treatment cards for remote consultation before the interview (the patients who met the criteria). 2. Informed consent (all patients). 3. Lost time and costs before the current antituberculosis treatment (only for new cases interviewed during the intensive phase). 4. Lost time and costs during the TB episode, coping mechanisms, and social consequences (all patients).

## Approval of ethics committee

The approval process was backed by the Research Ethics and Methodologies Committee (CEMIN, in Spanish) of the National Health Institute of Colombia, with approval code No. CEMIN 48–2019. Verbal informed consents were recorded using the Windows Sound Recorder app and stored with a unique survey number before the interviews. Personal details that could identify participants in the study were encrypted and codified in compliance with the Habeas Data Law of Colombia. All parameters established in Resolution 8430 of 1993 were followed, thereby guaranteeing anonymity, and restricting personal details and the use and protection of personal identity [20].

## Analysis

**Data processing.** Data was stored as captured in a password protected database and availed on demand for preprocessing and analysis through a CSV format download. Data were reviewed to detect inconsistencies using advanced filters and cross tabulations in Microsoft® Excel, Tableau Prep (Tableau Software, LLC), and an accompanying documentation was used for data cleaning using STATA 17 (College Station, TX: StataCorp LLC). Subsequently,

analysis [21] was done guided by standardized analysis scripts accompanying the WHO handbook on patient cost surveys, generating the variables necessary to obtain the WHO standard results reporting formats, guiding the imputations of missing data, especially those relating to the hourly income of patients, and the cost extrapolations for people with DS and DR-TB throughout the expected treatment. The reported means and related confidence bounds are survey design weighted and cluster sampling adjusted using the R 4.0.1 statistic software (Comprehensive R Archive Network).

**Assessment of income.** Participants were asked about their individual and household income both before and after TB diagnosis [16]. Additionally, they reported about household assets and characteristics of the patient's dwelling, adapted from the National Administrative Department for Statistics' (DANE, in Spanish) 2019 Living Standard National Survey (ECV, in Spanish) [22]. These asset data were used to extrapolate household income of each respondent via a generalized linear model (GLM) with a gamma distribution. The results of this model were compared with the income distribution by deciles of the National Survey of Household Budgets (ENPH, in Spanish) [23], and with the self-reported income of each respondent in our survey to verify their correlation with what DANE had published in the ENPH.

**Estimation of the costs of a TB episode.** Total per-episode costs (direct medical, direct non-medical, and indirect) were estimated from the onset of symptoms to TB treatment completion. Direct medical costs were defined as any costs relating to clinic visits, medicine collection, administration of directly observed therapy, and hospitalization. Non-medical costs were those relating to transport, nutritional supplements, food, and accommodation. Indirect costs were computed as the loss in income during TB episode, commonly referred to as the output approach that computes changes in reported household income before TB, and at the time of the interview. The valuation of time spent on seeking tuberculosis care and treatment (the human capital approach [self-reported hourly wage] for indirect cost estimation) was applied in sensitivity analysis (S2 and S3 Tables) [16].

Retrospective costs and patient's loss of time were extrapolated within the treatment phase, outside of treatment, and until the finalization of the planned treatment [16]. This was done using median values (by type of TB) taken from information received from TB patients during the survey and considering the MSPS guideline which defines the framework for treatment in Colombia [24]. Total costs were expressed as a percentage of annual household income with a 95%CI. Costs and income were collected in national currency (Colombian pesos) and later converted using US$1 = COP 3.756,7 according to the conversion rate in 2021 [25].

**Estimation of catastrophic costs.** Each TB patient and their household was assigned a binary number dependent on whether they had or had not incurred CC associated to TB, as per WHO threshold of 20% of annual income [16]. These numbers allowed for calculation of the percentage of patients with TB and their households which incur CC related to the illness.

**Factors associated with tuberculosis costs.** To identify the factors that influence cost to TB patients and their households, the relation of the variables and the costs was explored by way of univariate and multivariate analysis. A stepwise logistic regression was performed using household and TB patient characteristics. A model selection was based on the Akaike Information Criterion (AIC); the odds-ratios of the selected logistic model were reported.

## Results

Table 1 shows clinical and sociodemographic characteristics of the study population. In total 1,065 participants were successfully reached, 1,035 (97.2%) with DS-TB and 30 (2.8%) with DR-TB. Of all participants surveyed, 55.9% were male (6) and 77.1% were in the continuation phase. 17.8% of participants were classified as extrapulmonary cases and 88.4% had a known

**Table 1. Demographic and clinical characteristics of people with TB surveyed in Colombia, 2021.**

| Demographic characteristics | DS-TB | DR-TB | Total |
|---|---|---|---|
| | N (%) | N (%) | N (%) |
| | **1.035 (100)** | **30 (100)** | **1.065 (100)** |
| **Sex** | | | |
| Female | 460 (44.4) | 10 (33.3) | 470 (44.1) |
| Male | 575 (55.6) | 20 (66.7) | 595 (55.9) |
| **Age group (years)** | | | |
| 18–24 | 142 (13.7) | 5 (16.7) | 147 (13.8) |
| 25–34 | 253 (24.5) | 7 (23.3) | 260 (24.4) |
| 35–44 | 168 (16.2) | 5 (16.7) | 173 (16.3) |
| 45–54 | 155 (15.0) | 2 (6.7) | 157 (14.8) |
| 55–64 | 153 (14.8) | 4 (13.3) | 157 (14.8) |
| ≥65 | 163 (15.8) | 7 (23.3) | 170 (16.0) |
| **Patient's education level** | | | |
| No education | 48 (4.6) | 1 (3.3) | 49 (4.6) |
| Primary school | 282 (27.2) | 8 (26.7) | 290 (27.2) |
| Secundary school or higher | 705 (68.1) | 21 (70.0) | 726 (68.2) |
| **Household size. Median (Min.–Max.)** | 4 (1–20) | 4 (1–13) | 4 (1–20) |
| **Reported monthly household income pre-TB, mean (95%CI)** | $392.0 (243.4–540.6) | $346.5 (323.7–369.3) | $347.7 (325.1–370.3) |
| **Treatment pase** | | | |
| Intensive | 236 (22.8) | 19 (63.3) | 255 (23.9) |
| Continuation | 799 (77.2) | 11 (36.7) | 810 (76.1) |
| **HIV status** | | | |
| Positive | 118 (11.4) | 2 (6.7) | 120 (11.3) |
| Negative | 797 (77.0) | 24 (80.0) | 821 (77.1) |
| Unknown | 120 (11.6) | 4 (13.3) | 124 (11.6) |
| **Type of TB** | | | |
| Pulmonary | 848 (81.9) | 27 (90.0) | 875 (82.2) |
| Extrapulmonary | 187 (18.1) | 3 (10.0) | 190 (17.8) |
| **Diagnostic delay (>4 weeks)** | 795 (76.8) | 14 (46.6) | 809 (75.9) |

and documented HIV status (6). Further demographic and clinical characteristics are presented in Table 1.

Approximately 10% of the participants surveyed reported at least one hospitalization event in their current treatment phase, with average hospitalization time being 21 days (95%CI 16.4–25.6 days). TB patients showed an average of 60 consultations during TB treatment and a delay of approximately two months for treatment initiation from the onset of symptoms. The characteristics of the model of clinical care for the participants are presented in Table 2.

The average cost of a TB episode in Colombia was US$1,218 (95%CI 1,106–1,330), with $860.9 (95%CI 776.1–945.7) coming by way of direct non-medical costs, $339.0 (95%CI 257–421) from indirect costs, and $18.1 (95%CI 11.9–24.4) for medical costs (Table 3) including 70.7% of direct non-medical costs, and 27.8% indirect costs. Indirect costs in DR-TB were 11.5 percentage points less than in DS-TB (S1 Fig). Most TB episode costs were attributed to post-diagnostic costs, particularly nutritional supplements ($384.9 [95%CI 318.6–451.3]) and transport/travel costs ($302.5 [95%CI 254.4–350.6]). Pre-diagnostic costs were considerably less than post-diagnostic costs in all contexts (Table 3).

Out of all respondents, 27.2 (95%CI 22.6–31.8) reported taking a loan or selling an asset in order to face the costs associated with the illness (Table 4). By income quintile, 36.2% (95%CI

**Table 2. Model of care for TB, treatment duration and treatment delay survey participants, Colombia, 2021.**

| | DR-TB | DS-TB | Total |
|---|---|---|---|
| | Mean (95%CI) | Mean (95%CI) | Mean (95%CI) |
| **Hospitalization** | | | |
| Hospitalized at the time of the interview, n (%) | 1 (3.3) | 16 (1.5) | 17 (1.6) |
| Hospitalized in the current phase, n (%) | 6 (20.0) | 95 (9.2) | 101 (9.5) |
| Days of hospitalization during the current phase | 8.3 (1.54–15.1) | 21.8 (16.9–26.6) | 21.0 (16.4–25.6) |
| **Ambulatory care** | | | |
| Number of consultations per episode: total | 67.0 (61.7–72.3) | 60.5 (56.5–64.5) | 60.6 (56.6–64.6) |
| Number of consultations: directly observed treatment | 71.6 (59.9–83.3) | 52.8 (45.6–60.0) | 53.4 (45.5–61.3) |
| Number of consultations: follow-up | 4.0 (3.2–5.0) | 3.5 (3.3–3.8) | 3.6 (3.3–3.8) |
| Number of consultations: dispensing of drugs | 28.1 (18.4–37.8) | 24.4 (18.9–30.0) | 24.4 (21.6–27.2) |
| Number of visits prior to diagnosis | 0.5 (0.2–0.8) | 0.2 (0.18–0.29) | 0.2 (0.19–0.31) |
| Number of consultations prior to diagnosis (private center) | 0.4 (0.31–0.68) | 0.2 (0.10–0.25) | 0.2 (0.11–0.24) |
| **Treatment duration** | | | |
| Duration of treatment: intensive phase, months | 5.9 (4.7–7.1) | 2.2 (2.1–2.3) | 2.3 (2.2–2.4) |
| Treatment duration: continuation phase, months | 7.7 (6.2–9.2) | 4.0 (3.9–4.1) | 4.1 (4.0–4.2) |
| **Days of treatment delay** | 94,0 (81,2–106,8) | 71,4 (67,0–75,8) | 71,9 (63,8–80,0) |

28.6–43.8) of those in the first quintile had asked for a loan, while in the wealthiest quintile this percentage was 17.9% (95%CI 12.4–23.4). We observed that as we move higher in the income categories, fewer patients take loans to confront costs during TB illness. The social consequences faced by TB patients in Colombia in 2021 include 26.7% (95%CI 21.1–32.4) of households suffering food insecurity; 3.0% (95%CI 1.6–4.4) divorcing; 44.1% (95%CI 38.5–49–7) losing their jobs; 4.1% (95%CI 2.9–5.3) seeing their studies interrupted; and 34.7% (95%CI 30.3–39.0) socially excluded. Only 8.3% of the participants surveyed reported having been unemployed before the onset of TB; during the TB illness unemployment increased to 39.2%, in which the majority came from informal jobs, as seen in Fig 1. Of those who responded that they had lost their jobs because of TB, the majority (65.0% [95%CI 59.6–70.4]) came from the Quintile 1, showing a gradient as income level rose (Table 4).

The percentage of households that experienced CC due to TB in Colombia was 51.7% (95% CI 45.4–58.0); sensitivity analyses are shown in S2 Fig. Comparing DS and DR-TB, the

**Table 3. Total costs incurred by TB-affected households during one TB episode in Colombia, 2021 (with indirect costs assessed using output approach in US$).**

| Cost category | | DS-TB | DR-TB | Total |
|---|---|---|---|---|
| | | Mean (95%CI) | Mean (95%CI) | Mean (95%CI) |
| **Pre-TB diagnosis** | Medical | 6.0 (1.8–10.2) | 21.6 (0–57.6) | 6.4 (2.1–10.8) |
| | Non-medical | 3.4 (0.4–6.3) | 15.6 (1.8–29.5) | 3.7 (1.0–6.4) |
| **Post-TB diagnosis** | Medical | 9.1 (4.3–13.9) | 101.6 (0–261.1) | 11.7 (6.0–17.4) |
| | Non-medical | 819.1 (738.2–900) | 2,185 (1,055–3,314) | 857.2 (772.9–941.5) |
| | Travel | 284.9 (244.3–325.5) | 915.1 (64.1–1,766) | 302.5 (254.4–350.6) |
| | Accommodation | 15.3 (0–33.1) | – | 14.8 (0–32.3) |
| | Food | 149.7 (112–187.3) | 184.9 (65.1–304.8) | 150.6 (112.5–188.8) |
| | Nutrition supplement | 365.4 (307.3–423.6) | 1,064 (286–1,843) | 384.9 (318.6–451.3) |
| **Subtotal** | Medical | 15.1 (9.3–20.9) | 123.2 (0–285) | 18.1 (11.9–24.4) |
| | Non-medical | 822.5 (740.8–904.2) | 2,200 (1,081–3,320) | 860.9 (776.1–945.7) |
| | Indirect costs (outcome approach) | 335.0 (247.8–422.1) | 479.9 (107.7–852.1) | 339 (257–421) |
| **Total** | | **1,173 (1.067–1.278)** | **2,803 (1,580–4,027)** | **1,218 (1,106–1,330)** |

**Table 4. Reported coping mechanisms and social consequences in households affected by TB in Colombia, 2021.**

| | Income quintiles | | | | | |
|---|---|---|---|---|---|---|
| | **POOREST** Quintile 1 % (95%CI) | **MODERATELY POOR** Quintile 2 % (95%CI) | **AVERAGE** Quintile 3 % (95%CI) | **MODERATELY WEALTHY** Quintile 4 % (95%CI) | **WEALTHIEST** Quintile 5 % (95%CI) | **ALL TB PATIENTS** % (95%CI) |
| **Coping mechanism and social consequences** | | | | | | |
| **Loans** | 36.2 (28.6–43.8) | 30.8 (24.4–37.1) | 38.4 (31.9–44.8) | 27.9 (22.2–33.6) | 17.9 (12.4–23.4) | 30.3 (26.4–34.2) |
| **Sale of assets** | 28.7 (22.8–34.7) | 20.4 (15.6–25.3) | 18.5 (13.9–23.1) | 15.2 (9.9–20.6) | 5.1 (2.3–7.9) | 17.8 (14.2–21.4) |
| **Loan or sale of assets** | 39.7 (32.2–47.2) | 28.6 (23.0–34.2) | 25.7 (18.5–32.9) | 29.7 (21.7–37.7) | 11.6 (7.2–16) | 27.2 (22.6–31.8) |
| **Food insecurity** | 43.3 (35.5–51.2) | 35.5 (26.0–45.1) | 24.6 (18.0–31.2) | 18.7 (11.5–25.9) | 9.6 (5.4–13.8) | 26.7 (21.1–32.4) |
| **Divorce/separation** | 4.4 (1.7–7.2) | 2.4 (0.2–4.7) | 1.6 (0.0–3.6) | 3.8 (1.4–6.3) | 2.6 (0.5–4.7) | 3.0 (1.6–4.4) |
| **Job loss** | 65.0 (59.6–70.4) | 55.5 (47.8–63.2) | 42.6 (34.6–50.6) | 34.3 (29.5–39.2) | 20.5 (15.8–25.2) | 44.1 (38.5–49.7) |
| **Interrupted schooling** | 3.8 (1.7–5.9) | 4.0 (1.6–6.5) | 7.0 (3.0–10.9) | 3.4 (1.2–5.7) | 2.2 (0.5–3.9) | 4.1 (2.9–5.3) |
| **Social exclusion** | 40.3 (32.6–48.0) | 36.8 (29.6–44.0) | 33.4 (28.0–38.7) | 33.1 (25.8–40.3) | 29.4 (23.0–35.8) | 34.7 (30.3–39.0) |
| **Self-reported impact (as a consequence of TB)** | | | | | | |
| **Richer** | 0.0 (0.0–0.0) | 0.0 (0.0–0.0) | 0.5 (0.0–1.5) | 0.0 (0.0–0.0) | 0.0 (0.0–0.0) | 0.1 (0.0–0.3) |
| **Equal** | 34.9 (28.0–41.8) | 39.1 (32.6–45.5) | 54.5 (47.2–61.8) | 59.8 (52.3–67.3) | 71.6 (66.2–77.1) | 51.5 (48.0–55.0) |
| **Poorer** | 48 (41.1–54.9) | 52.5 (45.1–60.0) | 38.1 (31.8–44.4) | 38.6 (31.8–45.4) | 27.4 (21.9–33.0) | 41.2 (37.9–44.5) |
| **Much poorer** | 17.2 (11.8–22.5) | 8.4 (4.2–12.6) | 6.9 (3.9–10.0) | 1.6 (0.2–2.9) | 0.9 (0.0–2.3) | 7.2 (5.7–8.6) |

CI: Confidence interval.

percentage of CC was less in the first group (51.3% [44.9–57.7] vs 65.0% [48.0–82.0]) (Table 5). CC analyzed only by direct medical costs in relation to annual income was 0,6% (95%CI 0.2–1.1) and 37.4% (95%CI 31.0–43.7) considering only direct medical and non-medical costs. CC by type of insurance showed that 46.2% (95%CI 40.3–52.1) of those in the contributive regime incurred costs above 20% of their annual household income, against 58.0% (95%CI 50.9–65.2) amongst those affiliated with the subsidized regime (poor population). The differences were more noteworthy when the threshold of CC was increased.

The percentage of households that faced CC due to TB in Colombia varied depending on the income quintile of the household (S1 Table). CC in households of the Quintile 1 was 70.5% (95%CI 63.6–77.4), while in the Quintile 5 was 25.1% (95%CI 19.8–30.3). By DS-TB, these percentages were 70.4% and 23.6%, respectively (S1 Table).

The multivariate analysis of the factors associated with CC due to TB is presented in the Fig 2. The income quintiles resulted statistically significant in the model; thereby as household income increased, the risk of incurring CC due TB was lower. Similarly, losing one's job made one 2.1 times more likely to incur CC than those who retained their jobs (OR adjusted: 2.1 95%CI 1.5–6.1). Additionally, having extra pulmonary TB and being a DR-TB patient augmented the risk of CC compared with having pulmonary TB (OR adjusted: 2.2, 95%CI 1.4–3.5) and with being a DS-TB patient (OR adjusted: 4.0, 95%CI 1.2–15.8), respectively (Fig 2).

## Discussion

This study is the first nationally survey to describe CC experienced by TB patients and their households in Colombia. It is the first survey at global level administered by phone, a novelty introduced as a strategy to adapt to the COVID-19 pandemic conditions which caused a decrease in outpatient care for patients of TB, where interviews are usually set as per WHO design. In addition to estimating one of the three indicators for the End TB strategy, we explored direct medical, non-medical, and indirect costs of a TB episode. Consistent with the country's reported notification data, 17.8% of participants were classified as extrapulmonary

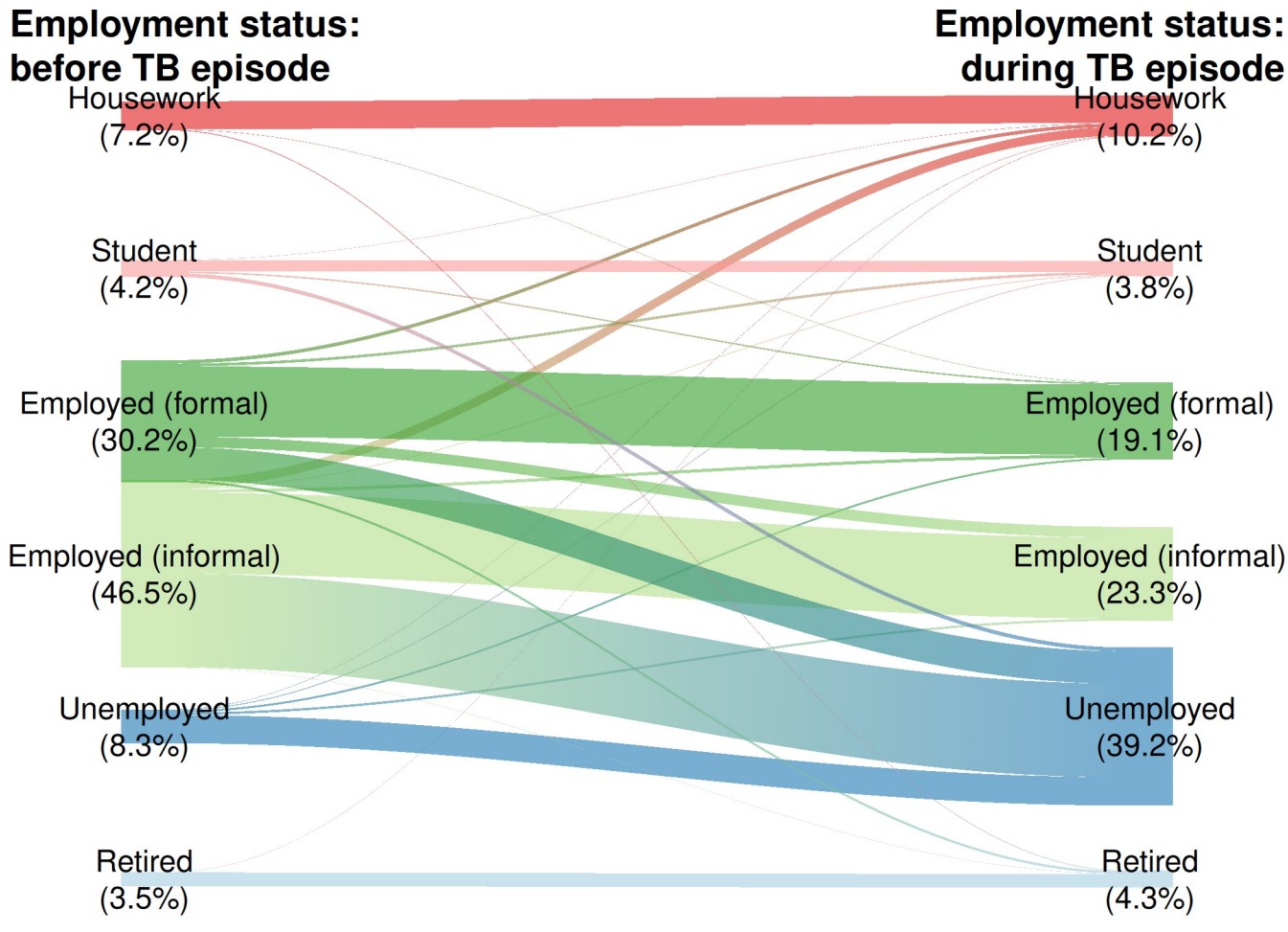

**Employment status: before TB episode**

Housework (7.2%)

Student (4.2%)

Employed (formal) (30.2%)

Employed (informal) (46.5%)

Unemployed (8.3%)

Retired (3.5%)

**Employment status: during TB episode**

Housework (10.2%)

Student (3.8%)

Employed (formal) (19.1%)

Employed (informal) (23.3%)

Unemployed (39.2%)

Retired (4.3%)

**Fig 1. Changes in the work situation before and after an episode of tuberculosis in Colombia, 2021.**

cases (national proportion in 2021 was 16.1%) and 88.4% had a known and documented HIV status (national proportion in 2021 was 90%) [6]. Our results depicted that in Colombia a substantial percentage of households (51.7% [95%CI 45.4–58.0]) face CC owing to TB. This percentage was higher in poor people (affiliated to subsidized regime) (58.0% [95%CI 50.9–65.1]) than with those on the contributive regime (46.2% [95%CI 40.3–52.1]). Colombia has a 99% coverage rate amongst its population, however, the difference in the CC by type of health coverage can be seen as correlative to socioeconomic status [14]. Also, CC associated to TB affected the poorest quintile (Quintile 1) disproportionately compared with Quintile 5 (highest income), showing a ratio of 2.8. Globally, this ratio was estimated in ∼1.8 by Portnoy et al. [26]. In Vietnam the ratio between quintiles 1 and 5 was 2.0 [27], showing a worrying inequality in Colombian households affected by TB. By DS-TB, this ratio was ∼3 times.

Our results are consistent with surveys taken in 29 other countries, where the pooled average weighted for each country's number of notified cases was 49% (95% CI: 37–61%) [10]. For Colombia, the indicator varied between 47.0–51.7%, depending on the methodology of human capital (S2 and S3 Tables) or the output approach. CC for TB treatment in Colombia are similar to the Brazilian estimates [28] and higher than in Fiji, the other countries with medium to high income level that finished the survey. Additionally, Colombia's value for CC due to TB was inferior to estimates taken in Burkina Faso, the Democratic Republic of Congo, Myanmar,

**Table 5. Proportion of household that face catastrophic costs due to TB according to different thresholds and scope of costs considered*.**

| Threshold | DS–TB | DR–TB | Total |
|---|---|---|---|
| **Proportion of households experiencing direct medical costs above various thresholds of annual income** | | | |
| 20% | 0.5% (0.1–0.9) | 6.2% (0.9–11.4) | 0.6% (0.2–1.1) |
| 30% | 0.4% (0.1–0.7) | 6.2% (0.9–11.4) | 0.5% (0.1–0.9) |
| 40% | 0.2% (0–0.4) | 6.2% (0.9–11.4) | 0.4% (0–0.7) |
| 50% | 0.1% (0.0–0.3) | 3.4% (0.0–7.5) | 0.2% (0–0.4) |
| 60% | 0.1% (0.0–0.3) | – | 0.1% (0.0–0.3) |
| **Proportion of households experiencing direct medical and non–medical costs above various thresholds of annual income** | | | |
| 20% | 36.8% (30.3–43.3) | 57.7% (42.1–73.3) | 37.4% (31.0–43.7) |
| 30% | 25.7% (20.2–31.2) | 54.9% (42.5–67.3) | 26.5% (21.3–31.8) |
| 40% | 19.1% (14.5–23.8) | 51.5% (36.0–67.0) | 20% (15.6–24.5) |
| 50% | 15.6% (11.4–19.9) | 33.2% (19.0–47.3) | 16.1% (12.2–20.0) |
| 60% | 11.4% (7.8–15.0) | 29.8% (16.9–42.6) | 11.9% (8.5–15.3) |
| **Proportion of households experiencing direct medical, non–medical and indirect costs above various thresholds of annual income** | | | |
| 20% | 51.3% (44.9–57.7) | 65% (48.0–82.0) | 51.7% (45.4–58) |
| 30% | 37.9% (31.5–44.4) | 62.2% (47.4–77.0) | 38.6% (32.5–44.8) |
| 40% | 25.6% (20.2–31.1) | 58.8% (41.2–76.3) | 26.6% (21.4–31.7) |
| 50% | 19.5% (14.5–24.6) | 40.5% (28.0–53.0) | 20.1% (15.5–24.7) |
| 60% | 13.8% (9.7–17.9) | 37% (26.0–48.1) | 14.5% (10.7–18.3) |
| **Proportion of households with a contributive regime experiencing direct medical, non–medical and indirect costs above various thresholds of annual income** | | | |
| 20% | 45.5% (39.4–51.7) | 66.3% (41.6–91.1) | 46.2% (40.3–52.1) |
| 30% | 31.9% (26.7–37.1) | 60.7% (41.9–79.5) | 32.9% (28.0–37.7) |
| 40% | 19.1% (15.9–22.2) | 60.7% (41.9–79.5) | 20.4% (17.4–23.4) |
| 50% | 12.9% (9.3–16.6) | 31.1% (0.0–67.2) | 13.5% (10.5–16.6) |
| 60% | 8.5% (5.9–11.0) | 31.1% (0.0–67.2) | 9.2% (7.0–11.5) |
| **Proportion of households with a subsidized regime experiencing direct medical, non–medical and indirect costs above various thresholds of annual income** | | | |
| 20% | 58% (50.9–65) | 57.2% (39.1–75.2) | 58.0% (50.9–65.1) |
| 30% | 44.2% (36.7–51.6) | 57.2% (39.1–75.2) | 44.5% (37.2–51.7) |
| 40% | 31.4% (24.1–38.8) | 49.3% (26.7–71.9) | 31.9% (24.7–39.1) |
| 50% | 25.6% (19.3–31.9) | 40.4% (23.6–57.2) | 25.9% (19.9–31.9) |
| 60% | 18.4% (12.6–24.1) | 32.5% (13.2–51.8) | 18.7% (13.1–24.3) |

*Values in parenthesis are 95%CI.

Vietnam, Laos, Ghana, Mongolia, and others [10,27,29]. Worldwide, CC pooled average was 47% (95%CI: 35–59) for DS-TB and 83% (95%CI: 75–90) for DR-TB [8,10]. In Colombia these percentages were 51.7% and 65%, respectively. The percentage of patients with DR-TB and their households that suffered CC was 78.5% in Brazil [10,28].

Surveys also signaled the need for eliminating non-medical costs and mitigating loss of income [8,10]. Non-medical costs including transport/travel, food and nutritional supplements represent 70.7% of total costs in Colombia in 2021, surpassed only by the Solomon Islands (80%) [30] and Fiji (73%). Countries such as Kenya (54%) [31], Uganda (54%) [32], El Salvador (58%), Timor-Leste (53%) and the United Republic of Tanzania (51%) all reported lower proportions [10].

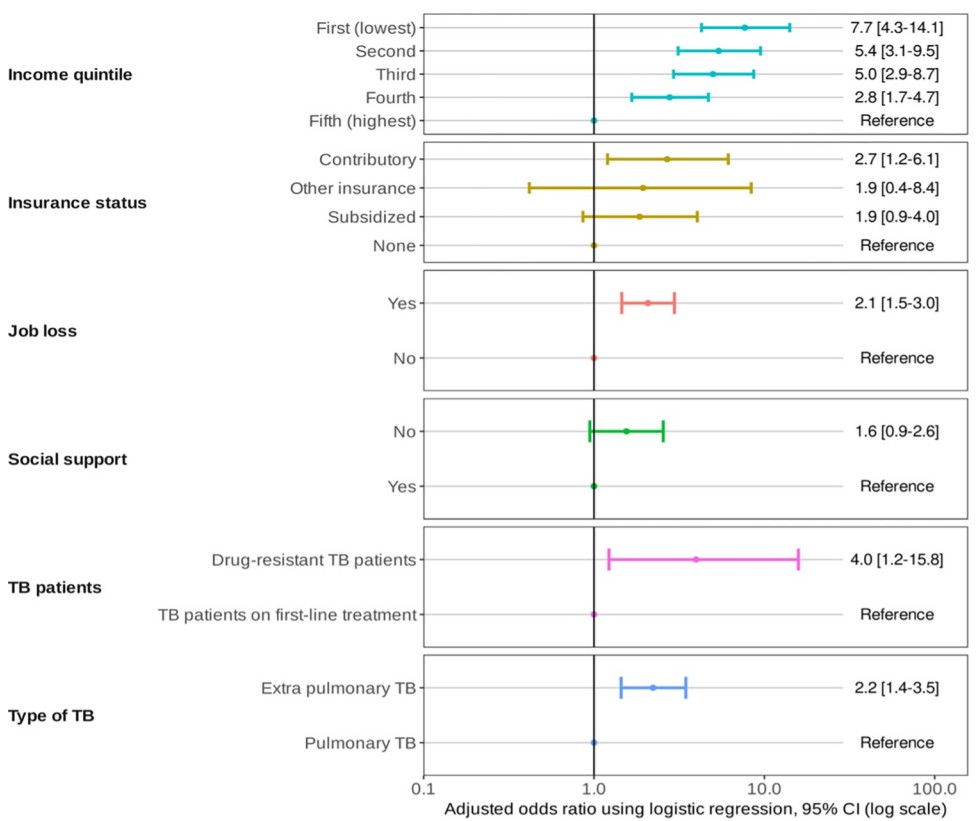

**Fig 2. Risk factors for TB-affected households facing costs >20% of household income or expenditure due to TB in Colombia, 2021.**

The average cost of an episode of tuberculosis in Colombia, from the onset of symptoms to the end of treatment, calculated based on the output approach, was US$ 1,218 (95%CI 1,106–1330). Looking at average total cost, DR-TB was ~2.4 times more costly for households than DS-TB. Also, it was 3.5 times greater than the average monthly salary of a household pre-TB; similar to Laos, where was more than 3 times the average monthly salary [33]. The calculated cost per episode in Colombia was also 5 times the monthly minimum wage in 2021 (US $241.8) [34]. TB costs estimated with alternative approach (sensitivity analysis), i.e. using human capital approach whereby time loss is multiplied by self-reported hourly wage were US $1,203 (95%CI 1,072–1,334)), 71.6% due to direct non-medical costs, 26.9% to indirect costs, and 1.5% to medical costs (S2 Table). If the estimated costs per episode were extrapolated to all the people with TB in Colombia, the economic burden for the households would be considerable.

Before TB, 76.1% of the people interviewed reported working and 8.3% reported being unemployed. During the TB episode unemployment increased to 38.9%, with the majority coming from informal jobs. This result should be taken with caution as it was presented in the context of the COVID-19 pandemic where the unemployment rate reached 21.4% in the March-May trimester of 2020, the highest in the recent Colombian economic history. During the time phone surveys were being taken the unemployment rate in Colombia was 13.1% [35]. It's probable that the people with TB that lost their jobs in the midst of strict lock-down [36,37], had not yet returned to work at the time of this survey's data collection (i.e. March 2021). Nevertheless, this is beyond the scope of this analysis.

TB patients and their households tried to confront the economic consequences of the illness in various ways. 27.2% of the affected patients reported taking a loan or selling assets to assume costs related to TB (39.7% among the poorest surveyed- Quintile 1-). Twenty-six point seven percent of the surveyed patients reported having experienced food insecurity (4.5 times greater in quintile 1 vs. quintile 5), like results obtained in the Social Pulse Survey conducted by DANE, which shows that in the 23 principal cities in Colombia 30% of people do not eat three daily meals [38] and 44.1% claimed they had lost their jobs (job loss was 3 times more likely in Quintile 1 vs. Quintile 5). Additionally, 34.7% felt socially excluded because of the illness, while 48.4% felt poor or poorer than before the diagnosis of the illness, similar to the poverty index (*Incidencia de pobreza monetaria*) reported by DANE in Colombia (42.2%) [39].

Amongst the variables associated with the probability of facing CC from TB the survey found that losing one's job and the wealth quintile increased the risk of CC. The type of TB also factored into the incidence of CC as those with pulmonary cases suffered far less economically than those with extrapulmonary TB. These risk factors were associated with CC in previous surveys in Myanmar [40], Laos [33] y Vietnam [27].

Our study has some limitations. In logistical terms, CC due to TB estimates did not consider several vulnerable groups like homeless, incarcerated, and people younger than 18 years (the latter usually included in National TB patient cost surveys following WHO design). These groups are difficult to follow-up and legal and national ethic review board constraints did not allow their inclusion in this study, as occurred in other Latin-American countries [41]. On the other hand, some municipalities presented limitations in finding sufficient cases which met the criteria to be included in the study. This led us to generate replacement cases, something that is common in these types of studies. For this study, the sampling framework consisted of those with TB who had been notified, and only those with diagnoses registered by the PNPCT. Due to the limited costing horizon of the study (only till the end of the current TB episode's treatment), there was no way to describe and cost the impact of persistent disability due to TB [42]. The estimate for the cost per episode of TB assumed that the person with TB would finish the treatment. Estimates were made based on where in the respective treatment each patient was, and then extrapolating to the other phases of treatment based on median estimated costs in other patients who were interviewed [16]. This permitted us to reconstruct the cost per episode of each person, regardless of whether they had reported costs for each phase of treatment. Assuming that patients finish treatment is a limitation as it doesn't account for patients who may not finish treatment because of system related loss to follow-up (14). A longitudinal study design, as has been recently done in India [43] provides an alternative that can provide insight onto these important aspects of patient care and wellbeing, but such survey has larger time and financial investments requirements.

The results obtained in this study will serve as input in monitoring efforts to eliminate CC to patients with TB, as well as serving as a baseline for future cost surveys for other infectious diseases in Colombia. TB is an illness that has been prioritized through programs and actions of interest in public health, as stated in the Ten-Year Public Health Plan [44], the Strategic Plan Towards the End of TB 2016–2025 [11], and the Resolution 227 of 2020 [24]. Also, the Colombian Congress enacted a law which will guarantee care for those with tuberculosis [45]. All of these establish the guidelines for prevention, diagnosis, treatment, and integral attention for people with TB in accordance with the latest directives from the WHO. This study showed that the direct medical costs due to TB were relatively low in comparison with other types of costs. Nevertheless, a substantial percentage of non-medical and indirect costs causing loss of income merit intersectoral and multi-sectoral oversight with other ministries at the national level, in order to help protect affected households from these CC. Specifically, costs associated with travel, nutrition supplements, additional food, the removal of (or a reduction in) any one

of the beforementioned costs would lower the economic burden imposed on TB patients in Colombia. Facilitating access to existing social protection schemes, legislation that prevents layoffs for the sick, territorialization, integrality, intersectionality, social participation (hearing community voice), even food/transport vouchers surely would reduce the household financial implications of the disease in Colombia.

## Conclusion

In conclusion, the results of our survey showed that although TB treatment is majorly financed by the Colombian healthcare system, the income losses (before and during a TB episode) caused by high non-medical and indirect costs continue to be a major problem for households of patients. High costs for low-income TB patients elevate the barrier of access to and put into danger the adherence to current TB treatments. Our results suggest the need for expanding the existing social programs that permit the reduction in the percentage of TB households incurring CC, especially those from low-income quintiles.

## Supporting information

**S1 Checklist.**
(DOCX)

**S1 Fig. Composition of TB patient costs in Colombia by drug resistant status, 2021.**
(TIF)

**S2 Fig. Percentage of households affected by tuberculosis that incur catastrophic costs with different thresholds (20% and others) in Colombia, 2021.**
(TIF)

**S1 Table. Proportion of household that face catastrophic costs due to TB (output approach) according to different thresholds, type of costs, and quintiles of wealth, Colombia, 2021.** *Not estimated due to low sample size.
(DOCX)

**S2 Table. Total costs incurred by TB-affected households during one TB episode in Colombia, 2021, assessed by human capital method (in US$).** In parenthesis 95% confidence intervals.
(DOCX)

**S3 Table. Proportion of household that face catastrophic costs due to TB according to different thresholds and type of costs (human capital method).**
(DOCX)

## Acknowledgments

We would like to thank interviewers, regional and local TB referents for their support conducting the interviews.

## Author Contributions

**Conceptualization:** Ingrid García, Ernesto Montoro, Peter Nguhiu, Inés García Baena.

**Data curation:** Nelson J. Alvis-Zakzuk, Juan Carlos Moreno Cubides, Ángela María Sánchez Salazar, Peter Nguhiu, Inés García Baena.

**Formal analysis:** Nelson J. Alvis-Zakzuk, Juan Carlos Moreno Cubides, Ángela María Sánchez Salazar, Peter Nguhiu.

**Funding acquisition:** Oscar Andrés Cruz Martínez, Gloria Mercedes Puerto, Julián Trujillo Trujillo, Claudia Marcela Castro-Osorio, Carlos Castañeda-Orjuela, Inés García Baena.

**Methodology:** Oscar Andrés Cruz Martínez, Ingrid García, Nelson J. Alvis-Zakzuk, Martha Patricia López, Juan Carlos Moreno Cubides, Ángela María Sánchez Salazar, Peter Nguhiu, Inés García Baena.

**Project administration:** Oscar Andrés Cruz Martínez, Ingrid García, Gloria Mercedes Puerto, Martha Patricia López, Julián Trujillo Trujillo, Claudia Marcela Castro-Osorio, Vivian Vanessa Rubio, Carlos Castañeda-Orjuela, Ernesto Montoro, Inés García Baena.

**Supervision:** Oscar Andrés Cruz Martínez, Gloria Mercedes Puerto, Martha Patricia López, Juan Carlos Moreno Cubides, Claudia Marcela Castro-Osorio, Vivian Vanessa Rubio, Ernesto Montoro.

**Validation:** Nelson J. Alvis-Zakzuk, Peter Nguhiu, Inés García Baena.

**Visualization:** Nelson J. Alvis-Zakzuk, Peter Nguhiu, Inés García Baena.

**Writing – original draft:** Nelson J. Alvis-Zakzuk, Inés García Baena.

**Writing – review & editing:** Oscar Andrés Cruz Martínez, Ingrid García, Gloria Mercedes Puerto, Nelson J. Alvis-Zakzuk, Martha Patricia López, Juan Carlos Moreno Cubides, Ángela María Sánchez Salazar, Julián Trujillo Trujillo, Claudia Marcela Castro-Osorio, Vivian Vanessa Rubio, Carlos Castañeda-Orjuela, Ernesto Montoro, Peter Nguhiu.

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
