## [Decision Letter · Decision Letter 0]

18 Jul 2023

PONE-D-23-12324Tuberculosis patients face high treatment support costs in Colombia, 2021PLOS ONE

Dear Dr. Alviz-Zazuk,

Thank you for submitting your manuscript to PLOS ONE. After careful consideration, we feel that it has merit but does not fully meet PLOS ONE’s publication criteria as it currently stands. Therefore, we invite you to submit a revised version of the manuscript that addresses the points raised during the review process.

ACADEMIC EDITOR:The manuscript is well written and the findings will benefit TB programs not only in Colombia but in several countries across the world. Before submitting your revised version ensure that all tables, figures and references are align with plosone publication guidelines - https://journals.plos.org/plosone/s/submission-guidelines  

Please submit your revised manuscript by Sep 01 2023 11:59PM. If you will need more time than this to complete your revisions, please reply to this message or contact the journal office at plosone@plos.org. Please include the following items when submitting your revised manuscript:A rebuttal letter that responds to each point raised by the academic editor and reviewer(s). You should upload this letter as a separate file labeled 'Response to Reviewers'.A marked-up copy of your manuscript that highlights changes made to the original version. You should upload this as a separate file labeled 'Revised Manuscript with Track Changes'.An unmarked version of your revised paper without tracked changes. You should upload this as a separate file labeled 'Manuscript'.If applicable, we recommend that you deposit your laboratory protocols in protocols.io to enhance the reproducibility of your results. Protocols.io assigns your protocol its own identifier (DOI) so that it can be cited independently in the future. For instructions see: https://journals.plos.org/plosone/s/submission-guidelines#loc-laboratory-protocols. Additionally, PLOS ONE offers an option for publishing peer-reviewed Lab Protocol articles, which describe protocols hosted on protocols.io. Read more information on sharing protocols at https://plos.org/protocols?utm_medium=editorial-email&utm_source=authorletters&utm_campaign=protocols.

We look forward to receiving your revised manuscript.

Kind regards,

Ibrahim Jahun, MD, MSC, PhD

Academic Editor

PLOS ONE

Journal Requirements:

"YES. the following institution funded this study:

World Health Organization

Panamerican Health Organization

Ministerio de Salud y Protección Social

Instituto Nacional de Salud "      

"NO authors have competing interests"

Reviewers' comments:

Reviewer's Responses to Questions

**Comments to the Author**

1. Is the manuscript technically sound, and do the data support the conclusions?

Reviewer #1: Yes

Reviewer #2: Yes

2. Has the statistical analysis been performed appropriately and rigorously? 

Reviewer #1: Yes

Reviewer #2: Yes

3. Have the authors made all data underlying the findings in their manuscript fully available?

Reviewer #1: No

Reviewer #2: Yes

4. Is the manuscript presented in an intelligible fashion and written in standard English?

Reviewer #1: Yes

Reviewer #2: Yes

5. Review Comments to the Author

Reviewer #1: The manuscript is well written. The findings are interesting and would go along way in supporting policymakers to address TB situation in Colombia. Please address the following observations:

1. There are several typos, please review and address them. The paper is not line numbered and I therefore cannot provide specific line references to ease your edit.

2. It will be good to add some socioeconomic characteristics of the households because these have direct relationship with CC among the TB patients. Alternatively, consider adding table 1s in the body of the paper. This table is very important.

3. In the result section, language should be revised to ensure results are just stated and not discussed. Some sentences in the result section are more like “discussion”.

4. It will be good to indicate how CC is distributed among 32 departments and Bogota. This level of details may be useful for policy makers to design department-based interventions thereby ensuring efficiency in resource allocation.

5. In table 1, HIV is indicated as clinical factors but this was not discussed. We all know that HIV is critical comorbidity among TB patients and obviously this may serve as a confounder thereby affecting CC. Additionally, the authors may need to explain how other confounders such as other chronic illnesses were addressed. Perhaps these could be attributed to limitations of the study.

Reviewer #2: Reviewer Comments

General comments

1. Overall, this is a seminal and innovative work carried out by the authors exploring a very important aspect of care for TB patients primarily, with potential implication for other diseases.

2. There are no significant issues related to methodology and overall approach.

3. The authors should proof read and pay attention to language, grammar, syntax, brevity, formatting/spacing, and typos

4. The manuscript could benefit from further review by a native English speaker.

Abstract

1. Consider moving the definition of catastrophic cost to the methods section.

2. The last part of the last sentence of the methods section should simply read: ‘analyzed for determinants/risk factors of CC.’

3. The last sentence of the conclusion should either be specific with the categories of those at risk of CC rather the generalization: ‘vulnerable’ since there was no definition of vulnerable in the main manuscript or abstract. The conclusion section in the main manuscript is much better in this regard.

Main manuscript

Introduction

1. For brevity, consider removing confidence intervals in the rates reported. The reader may go to the relevant publications for the details.

2. Please clarify if the sentence: ‘Moreover, the PNPCT reports that only 71% of cases under treatment in 2020 achieved successful treatment outcomes against a target of 90%, and 10% of cases were lost to follow-up during the treatment process’ means 10% is also a target rather than an actual measure.

3. In the last sentence, consider deleting ‘cost of TB’ from the objectives as it is part of many variables in the study that aimed towards estimating CC and its determinants.

Materials and Methods

1. In the ‘Sampling’ sub-section state how individuals were selected from each of the 27 clusters. Did the authors include all the eligible subjects in the selected clusters or use some probability sampling method? Please be more explicit.

Results

1. Table 1 is too ‘busy’. Consider deleting the percentage columns and place the percentages in brackets beside the whole numbers as was done with the 95% CI in Table 2.

2. Move the comparison of rates and proportion with the study findings to the discussion section. Limit the result section to just the results.

3. For table 3, same comments applies as for Table 1 in (1) above.

Discussion

1. Limit the tendency to repeat the presentation of results in the discussion section except where it is necessary (such as comparisons).

Limitations

1. A lot of vulnerable groups were excluded based on the said ethical considerations. How this will affect the findings and conclusions therefrom should be highlighted in the limitation section.

6. PLOS authors have the option to publish the peer review history of their article (what does this mean?). If published, this will include your full peer review and any attached files.

Reviewer #1: No

Reviewer #2: **Yes: **Mahmood Dalhat

---

## [Author Response · Author response to Decision Letter 0]

3 Nov 2023

We atteched the response letter with all the corrections made.

---

## [Decision Letter · Decision Letter 1]

10 Dec 2023

Tuberculosis patients face high treatment support costs in Colombia, 2021

PONE-D-23-12324R1

Dear Dr. Alviz-Zazuk,

We’re pleased to inform you that your manuscript has been judged scientifically suitable for publication and will be formally accepted for publication once it meets all outstanding technical requirements.

Kind regards,

Ibrahim Jahun, MD, MSC, PhD

Academic Editor

PLOS ONE

Additional Editor Comments (optional):

Reviewers' comments:

Reviewer's Responses to Questions

**Comments to the Author**

1. If the authors have adequately addressed your comments raised in a previous round of review and you feel that this manuscript is now acceptable for publication, you may indicate that here to bypass the “Comments to the Author” section, enter your conflict of interest statement in the “Confidential to Editor” section, and submit your "Accept" recommendation.

Reviewer #1: All comments have been addressed

Reviewer #2: All comments have been addressed

2. Is the manuscript technically sound, and do the data support the conclusions?

Reviewer #1: Yes

Reviewer #2: Yes

3. Has the statistical analysis been performed appropriately and rigorously? 

Reviewer #1: Yes

Reviewer #2: Yes

4. Have the authors made all data underlying the findings in their manuscript fully available?

Reviewer #1: Yes

Reviewer #2: Yes

5. Is the manuscript presented in an intelligible fashion and written in standard English?

Reviewer #1: Yes

Reviewer #2: Yes

6. Review Comments to the Author

Reviewer #1: The authors have improved the paper by responding to the comments provided. Results and Discussion sections are now in agreement.

Reviewer #2: (No Response)

7. PLOS authors have the option to publish the peer review history of their article (what does this mean?). If published, this will include your full peer review and any attached files.

Reviewer #1: No

Reviewer #2: **Yes: **Mahmood Muazu Dalhat

---

## [Editor Report · Acceptance letter]

29 Mar 2024

PONE-D-23-12324R1 

PLOS ONE

Dear Dr. Alvis-Zakzuk, 

I'm pleased to inform you that your manuscript has been deemed suitable for publication in PLOS ONE. Congratulations! Your manuscript is now being handed over to our production team.

Kind regards, 

on behalf of

Dr. Ibrahim Jahun 

Academic Editor

PLOS ONE